# Accounting for China's Net Carbon Emissions and Research on the Realization Path of Carbon Neutralization Based on Ecosystem Carbon Sinks

**Nuo Wang, Yuxiang Zhao, Tao Song \*, Xinling Zou, Erdan Wang and Shuai Du**

School of Economics and Resource Management, Beijing Normal University, Beijing 100875, China
\* Correspondence: songtao@bnu.edu.cn

**Abstract:** Carbon sinks are an important way to achieve carbon neutrality. In this study, carbon emissions in each year from 2019 to 2060 were predicted by constructing the LEAP (Long-range Energy Alternatives Planning System)-China model. The ecosystem carbon sinks in five representative years of 2012, 2017, 2019, 2030, and 2060 were predicted by reviewing related literature to calculate China's net carbon emission accounts in these five key years and to quantitatively analyze the path to achieving carbon neutrality in China. The results show that China's annual carbon emissions will peak in 2028, with a peak of 10.27 billion tons of carbon dioxide; that they will then decrease year by year to 7227 million tons of carbon dioxide in 2060; and that the ecosystem carbon sinks generated by land use are more stable, with a total of approximately 5.5 billion tons of carbon dioxide. To achieve carbon neutrality, a dependence only on ecosystem carbon sinks is insufficient. National energy conservation, voluntary emission reduction by enterprises, and a reliance on new energy and new technologies are needed to ensure the final implementation of China's carbon neutrality strategy.

**Keywords:** carbon neutral; carbon sink; NEP; LEAP-China; carbon peak



## 1. Introduction

With the continuous and rapid development of industrialization and urbanization, human activities have caused a dramatic increase in carbon emissions. A large amount of greenhouse gas emissions has caused global warming. On 22 September 2020, at the 75th session of the United Nations General Assembly, the Chinese government proposed that China will increase its independent national contribution, adopt more vigorous policies and measures, strive to reach the peak of carbon dioxide emissions by 2030, and strive to achieve carbon neutrality by 2060. Carbon neutrality is the new stage after carbon peaking and the key stage pursued by low carbon development. Carbon neutrality is defined as the total amount of greenhouse gas emissions generated by human activities within a certain period, offsetting the greenhouse gas emissions generated through ecosystem carbon sinks, energy conservation, and emission reduction and achieving zero greenhouse gas emissions. As a major carbon-emitting country, China has global influence in the field of global climate governance. Achieving carbon neutrality will contribute to the sustainable and healthy development of China's economy in the future and provide a guarantee for the sustainable development of all human civilizations.

The carbon sink refers to the process by which forests, grasslands, farmland, wetlands, and other ecosystems reduce carbon emissions by absorbing carbon dioxide. Approximately 2.8–18.7% of carbon emissions can be neutralized in 2060 through terrestrial ecosystem carbon sinks alone without policy intervention [1]. China is a vast country with rich plant species and well-functioning ecosystems, which have considerable carbon sink potential. Compared with other emission reduction methods, increasing the ecosystem carbon sink has a lower cost and is an easier method. Increasing ecosystem carbon sinks not only brings people closer to carbon neutrality in terms of numbers, but also brings changes to

the ecological environment in real life, thus improving people's quality of life. Therefore, we should pay more attention to ecosystem carbon sequestration in the path of carbon neutralization.

The existing "carbon neutralization" studies rarely consider an ecosystem carbon sink, while most studies considering ecosystem carbon sinks are based on historical data. Few people associate a carbon sink with the carbon neutralization in the future. Through the accounting for carbon sinks and carbon emissions, this study calculates the net carbon emission account of China in each key year. Through authoritative carbon account data, this study clearly and accurately describes the path to achieving carbon neutralization. Combined with specific values, we can determine the best path for China to achieve carbon neutralization from both ends of "sink and source". It fills the gap in the academic community of quantitative analysis of carbon neutral pathways based on a carbon sink perspective.

## 2. Literature Review

### 2.1. Literature on Carbon Neutralization

Carbon neutrality is a systematic project [2]. In addition to relying on ecosystem carbon sinks, the introduction of new energy saving and emission reduction technologies and the use of new energy can facilitate carbon neutralization. The realization of carbon neutralization is inseparable from the joint efforts of various fields. Some scholars [3] have described the pathways to carbon neutrality in different areas based on the current situation and policy planning [3–5]. Some scholars have also studied the contribution of technologies, such as negative emission technologies for buildings, CCUS, and BECCS to achieve carbon neutrality [6,7]. Other scholars take a social science perspective and analyzed the new situation of cultural industry development in the context of "double carbon" [8], proposed the path and suggestions for China's energy transition under the dual carbon target [9], and based on the objective actual situation, proposed the three-stage/four-step strategy and specific suggestions for China to achieve carbon neutrality [10].

On a regional scale, scholars have analyzed the pathways to carbon neutrality at all levels of government using different methods. These include estimating Mexico's achievement of a carbon neutrality target in 2050 through four simulated energy conversion scenarios [11]; and calculating the ecological cost of achieving the carbon neutrality target under different models based on carbon accounting theory, which provides a reference for carbon deficit provinces to choose carbon neutrality achievement paths [12]. Some scholars conducted a 25 species scenario analysis of the zero-carbon city construction path of Xiong'an New Area based on the LEAP model, thus providing effective suggestions for carbon neutrality in this region [13]. Some scholars assessed the carbon accounts of Shenzhen from 2000 to 2008 based on carbon source/sink coefficients and traffic flow prediction models [14]. Other scholars analyzed the spatial distribution of carbon metabolism in Beijing using GIS software and empirical coefficients [3].

In terms of the national strategic context, carbon neutrality is a necessary national-level development plan. On the issue of carbon neutrality in China, most of the relevant literature has proposed suggestions for China to achieve the goal of carbon neutrality from a certain perspective [15,16]. There are also a few studies that quantify the carbon neutrality target from the perspective of carbon sources and sinks and map out a path to achieving carbon neutrality in China. For example, Huang mapped out China's carbon-neutral pathway by predicting the future total carbon emissions and carbon sinks in China [17]. Through the independently developed national energy economy model, Yu investigated China's medium- and long-term $CO_2$ emission targets and realization paths and analyzed the carbon neutralization progress under different scenarios of economic growth and emission reduction [18].

### 2.2. Literature on Carbon Emission Calculation

The calculation of carbon emissions is technically difficult, and certain scholars use various sophisticated instruments to detect the emission sources or to quantify the greenhouse gas emissions from data collected by weather stations, while other scholars use models to estimate the carbon emissions in a region from data such as nighttime lights [19]. In addition, the choice of the extent of carbon emissions is also a key concern of the study because, in addition to energy activities, human and animal respiration also emit carbon dioxide. Since the carbon dioxide to be neutralized in carbon neutralization refers only to anthropogenic carbon dioxide, the carbon emissions calculated in this study only consider the carbon dioxide produced by energy consumption. Carbon dioxide produced by industrial processes, human and animal respiration, etc., were not considered.

The calculation of carbon emissions can be obtained directly with energy consumption through the carbon emission calculation method given by the IPCC or even quickly measured through tools (https://www.wri.org.cn/node/41204 (accessed on 28 January 2022.)) such as the Urban Greenhouse Gas Accounting Tool 2.2. However, these tools have high data requirements and cannot be utilized to make projections of future carbon emissions. Since energy consumption is highly correlated with the level of economic and social development [20] (the urbanization rate, population size, industrial structure, economic level of cities, energy intensity, energy structure, etc.), these are key factors affecting carbon emissions. Current authoritative, carbon emission measurement models include the Kaya model, the LMDI decomposition method, the emission inventory method, the CGE model, and the IPAT model [21]. Many domestic scholars' models for measuring carbon emissions are based on the improvement and extension of the these models, such as a generic carbon emission model that is based on the production function theory [20]. The IPAT model removes the population factor and introduces the variable labor rate of return, which can characterize the industrial structure and the level of science and technology [22]. Certain scholars have also employed other self-researched models for carbon emission forecasting, such as the self-developed [3]CIAM/NET model, to simulate China's future net carbon emission path [18]. The LEAP model was used to simulate the carbon peak and carbon neutralization path of the Xiongan new district under different scenarios [13].

### 2.3. Literature on Carbon Sink Calculation

The standards for carbon sink calculations are not uniform among academics, therefore the results of carbon sink calculations usually vary widely among scholars and differ in their methods. Some scholars have compiled and measured the carbon sequestration of various types of wetlands in China through literature review and data research and analysis, and have found significant differences between the findings of different scholars [23,24]. Fang estimated the carbon sink of terrestrial ecosystems in China using forest and grassland resource inventory information, agricultural statistics, and climate and other ground observations, as well as satellite remote sensing data, and by referring to the results of foreign studies [25]. The methods employed by the these scholars are ecological methods, and the results are calculated as the corresponding plant carbon sequestration or net first productivity in the criteria mentioned above. Since the ecological method requires consideration of various factors, such as vegetation area, plant species, and soil type, and is too difficult to operate. Many scholars have also estimated carbon sinks by using remote sensing data. For example, [19] used net primary productivity (NPP)/VIIRS remote sensing data to estimate the size of carbon sinks in counties in China. However, remote sensing data are usually NPP data, and the difference between them and net ecosystem productivity (NEP), which is the closest concept of carbon sink, is still large. In the literature on carbon offsetting, scholars usually use the net primary productivity of ecosystems [25–27] or the carbon sequestration rate [28]. NEP is conceptually the most consistent with the definition of a carbon sink, so we believe that it is most reasonable and feasible to use the NEP, which is widely used in academia, to calculate carbon sinks, by considering various aspects, such as operational difficulties and the relevance of the data.

### 3. Materials and Methods

Since the LEAP platform is an ideal tool for analyzing complex systems in the economic and energy environment [29] and is suitable for multi-scenario projections of carbon emissions at the national level over a long time horizon, this study constructed the LEAP-China model with the help of LEAP software to project carbon emissions over the period of 2020–2060, providing effective data support for carbon neutrality.

The process of carbon dioxide uptake and oxygen emission by ecosystems is complex, and four different criteria for calculating carbon sinks can be derived by considering different degrees of the complexity of this process. The first criterion is to consider the amount of carbon absorbed by photosynthesis as a carbon sink by considering only the carbon absorbed by green plants through photosynthesis and the oxygen emitted. However, since plants themselves also emit carbon through autotrophic respiration, the second criterion subtracts the carbon emissions from autotrophic respiration from the carbon absorbed by plants through photosynthesis to obtain the NPP. NEP is theoretically valid, but the actual ecosystem situation is complex and variable. Net biome production (NBP) can be obtained, which further estimates the long-term carbon sink [30]. The carbon sinks calculated in this study are province-based. In this study, the carbon sink is calculated by province, and the time is taken as one year. In view of the actual situation in space and time, the NEP was chosen as the standard for calculating the carbon sink.

Carbon emissions were predicted through the LEAP-China model, carbon sinks were measured through the NEP of land-use change, and the carbon account between the ecosystem carbon sink and energy consumption carbon emissions, which can reasonably and intuitively describe the path of carbon neutralization in China, was calculated. This approach is also a useful supplement to research on carbon neutralization in China.

### 3.1. Carbon Emissions Accounting and Forecasting

In this study, we constructed a LEAP-China model based on LEAP software to analyze and forecast $CO_2$ emissions based on energy consumption on the production side, which mainly includes energy consumption and $CO_2$ emissions accounting, and energy and $CO_2$ emissions forecasting based on socioeconomic conditions in China from the production side.

#### 3.1.1. Accounting for $CO_2$

The IPCC method was divided into two types: first, the sectoral method, which accounts for $CO_2$ emissions from different sectors based on their energy consumption, and the reference method, which accounts for $CO_2$ emissions from the country after summing them up; and second, the reference method, which accounts for $CO_2$ emissions from the country's total energy consumption, which is the difference among energy production, import and export, and inventory. The reference method accounts for $CO_2$ emissions based on the total national energy consumption, which is the difference among the amount of energy produced, imported, exported, and stocked. According to China's industrial sector classification and end-use energy consumption characteristics, this study adopted the IPCC sectoral method to account for China's fossil energy $CO_2$ emissions.

According to China's energy balance sheet, China's energy consumption sectors are divided into the primary industry (agriculture, forestry, animal husbandry, fishery, and water conservancy), secondary industry (industry and construction), tertiary industry (transportation, storage and post, telecommunications, wholesale and retail trade catering, and others), and residential life. Energy types are divided into four major categories: coal, oil, natural gas, and clean energy (i.e., nuclear energy and renewable energy). China's energy processing and conversion sector is divided into submodules, such as thermal power generation, oil refining, coking, heat supply, and gas production.

The energy demand In the end-consumption sector is:

$$SD_t = \sum_i AL_{ti} * EL_{ti}$$

where $SD_t$ is the end-use energy demand in sector t, $AL_{ti}$ is the activity level of subsector i in sector t, and $EL_{ti}$ is the corresponding energy consumption per unit activity level, i.e., energy intensity. Thus, for the end-consumption sector, the processing and transformation sector, and the calculation of $CO_2$ emissions associated with energy use, the following equations were used:

$$SE_t = \sum_i \sum_k SD_{tik} * EF_{tik}$$

$$TM_S = \sum_k L_{sk} * EF_{sk}$$

$$TE = \sum_t SE_t + \sum_s TM_s$$

where $SE_t$ is the $CO_2$ emissions from the end-consumption sector $t$; $SD_{tik}$ is the $k$-variety energy consumption in subsector $i$ of sector $t$; $EF_{tik}$ is the carbon emission factor corresponding to energy consumption; $TM_S$ is the $CO_2$ emissions from the process conversion sector $s$; $L_{sk}$ is the $k$-variety energy input for sector $s$; $EF_{sk}$ is the emission factor corresponding to energy consumption; and $TE$ is the total energy use-related $CO_2$ emissions.

China's carbon neutral target is greenhouse gas emissions from anthropogenic activities, so only carbon dioxide from energy consumption was considered in accounting for carbon emissions. In the LEAP-China model, the energy end-consumption sectors in China include all the sectors covered in the China Energy Balance Sheet and are divided into industrial and domestic sectors.

This study adopted the accounting method of carbon dioxide emissions formulated by the IPCC in 2006, where the industrial sector has the following equation:

$$C_1 = \sum_i \sum_j C_{ij} = \sum_i \sum_j \left( \frac{C_{ij}}{E_{ij}} * \frac{E_{ij}}{E_i} * \frac{E_i}{Y_i} * \frac{Y_i}{Y} * Y \right)$$
$$= \sum_i \sum_j CI_{ij} * ETS_{ij} * EI_i * ES_i * Y$$

where $i$ and $j$ denote industry type and energy type, respectively; $C_1$ denotes the total carbon dioxide emissions from the industrial sector; $C_{ij}$ denotes the carbon dioxide emissions from the $j$th fuel in industry $i$; $E_{ij}$ denotes the consumption of fuel $j$ in industry $i$; $E_i$ denotes the total energy consumption of industry $i$; $Y_i$ denotes the value added of industrial production in industry $i$; and Y denotes China's gross domestic product, or GDP. On the other hand, $CI_{ij}$ denotes the carbon emission factor of the $j$th fuel in industry $i$, $ETS_{ij}$ denotes the proportion of energy consumption of the $j$th fuel in industry $i$, $EI_i$ denotes the energy intensity of industry $i$, and $ES_i$ denotes the share of value added of production in industry $i$.

The subsistence sector is divided into urban and rural subsistence sectors, and the formula for total $CO_2$ emissions from the subsistence sector is expressed as follows:

$$C_2 = \sum_u \sum_v C_{uv} = \sum_u \sum_v CI_{uv} * ETS_{uv} * PI_u * TP_u$$

where $u$ denotes the urban or rural domestic sector, $v$ denotes the energy type, $C_2$ denotes the total carbon dioxide emissions from the living sector, $C_{uv}$ denotes the carbon dioxide emissions from the $v$th fuel in the $u$th living sector, $CI_{uv}$ denotes the carbon emission factor of fuel $v$ in sector $u$, $ETS_{uv}$ denotes the proportion of energy consumption of fuel $v$ in sector $u$, $PI_u$ denotes the energy intensity per capita in sector $u$, and $TP_u$ denotes the total population of the $u$th living sector.

### 3.1.2. LEAP Model Setting

The LEAP-China model has a base period of 2019 and a forecast period of 2020–2060. Based on the analysis above, China LEAP projections were divided into two parts: end-use energy consumption and process conversion. In the first part, $CO_2$ emissions from energy consumption were divided into two sectors, industrial consumption and residential

consumption, and the industrial sector was subdivided into primary, secondary and tertiary industries. For each sector, coal, oil, natural gas, and non-fossil energy consumption types were established.

The specific framework of the LEAP model is shown in Figure 1.

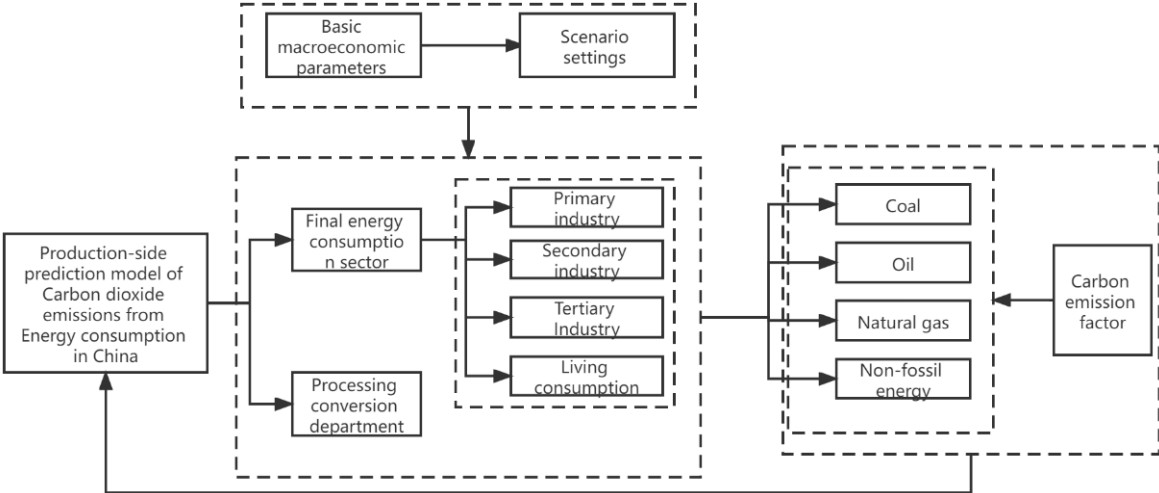

**Figure 1.** LEAP-China Model Framework.

### 3.1.3. Scenario Setting

Since economic and social development and the development and utilization of fossil energy are the main factors leading to carbon emissions, the drivers affecting long-term energy demand and $CO_2$ emissions in China can be divided into two parts: macroeconomic factors and policy factors. According to the research framework of the LEAP-China model and the need for scenario setting, the following indicators were selected as the main drivers of the model: carbon emission coefficient, energy consumption structure (i.e., the proportion of the different types of energy consumption), energy intensity, industrial structure, GDP, urbanization level (i.e., the proportion of the urban population), and population size. Referring to the relevant studies of existing scholars and historical data of developed countries in similar periods, the specific values of each parameter under a strong low-carbon scenario consistent with the national green development strategy were established as follows [31] Table 1.

**Table 1.** Key variable assumptions in LEAP-China.

| Meaning | 2019 | 2030 | 2060 |
|---|---|---|---|
| GDP growth rate/% | 6 | 3.92 | 2.0 |
| Percentage of primary industry/% | 7.5 | 6 | 3 |
| Percentage of secondary industry/% | 38.77 | 32 | 23 |
| Tertiary industry share/% | 53.77 | 62 | 78 |
| Population size/million people | 1410.08 | 1442 | 1393.4 |
| Urbanization rate/% | 62.71 | 70 | 85 |
| Share of clean energy generation/% | 32.7 | 42.4 | 65 |
| Primary industry energy intensity | 265.19 | 224.6 | 161.1 |
| Energy intensity of secondary industry | 1802.67 | 1218.2 | 768 |
| Tertiary sector energy intensity | 359.189 | 304.2 | 218.3 |
| Urban energy consumption per capita/ | 0.31 | 0.34 | 0.40 |
| Rural energy consumption per capita | 0.30 | 0.27 | 0.20 |

### 3.2. Carbon Sink Measurement

In this study, we used the NEP as a coefficient to calculate the regional carbon sink capacity with the following equation:

$$C_a = \sum S_i \theta_i + C_{aa}$$

where $S_i$ is the corresponding type of land area (five land types of forestlands, grassland, wetland, and garden land were selected in this study); $\theta_i$ is the NEP coefficient corresponding to the land type; and $C_{aa}$ is the carbon sink of agricultural crops. Since it is difficult to directly calculate the carbon sink of crops similar to other types of land, the carbon sink of crops was calculated separately by referring to the practice in related studies. Among them, wetlands include lake wetlands, river wetlands, and other waters, therefore five indicators of forestland, grassland, wetlands, garden land and crops were selected for the whole province.

Crop carbon sinks were referenced [32–36] practices, calculated using the following equation:

$$C_{aa} = \sum C_{aai} - \varphi * S_a = \sum C_{crop-i} \times (1 - P_{water-i}) \times \frac{Y_{eco-i}}{H_i} - \varphi * S_a$$

where $C_{aai}$ is the total annual NPP of a crop, $C_{crop-i}$ is the carbon uptake rate of the crop, $P_{water-i}$ is the water content of the crop, $Y_{eco-i}$ is the economic yield of the crop, and $H_i$ is the economic coefficient of the crop. Referring to research [37], the calculation results of the equation above correspond to crop NPP, and since the actual carbon sink to be used in this study was NEP, $C_{aai}$ was calculated. The heterotrophic respiration of the crop soil was subtracted from the base, where $\varphi$ is the soil heterotrophic respiration coefficient, and $S_a$ is the area of cultivated land. $C_{aa}$ is the calculated carbon sink of the farmland system.

NEP fluctuates greatly and has been shown to be influenced by various factors, such as climate, such that the maximum annual average coefficient of variation of NEP in China was 198% during the period from 1981 to 2000 [38]. Thus, the NEP fluctuation is not statistically significant [39]. Given the nature of NEP, it is extremely difficult to obtain a suitable and accurate NEP coefficient; thus, the NEP coefficients for forests and grasslands in this study were chosen from the most widely used studies in similar articles [40]. The parkland coefficient was determined by setting the NEP coefficient as the average of forest NEP and grassland NEP, in which the urban green area is unified into the garden area for calculation [41].

Because most of the soil in the wetland system is in an anaerobic environment for a long time, the decomposition of soil organic matter is slow [42], and periodic tides carry a large amount of $SO_4^{2-}$, which hinders the production of methane ($CH_4$) and reduces the generation and emission of carbon in salt marsh wetlands. Therefore, the heterotrophic respiration of wetland ecosystems will be smaller than that of other terrestrial ecosystems, and NPP can be used to approximately replace NEP. Since the carbon sequestration capacity of different wetlands varies greatly, the coefficients of wetlands were distinguished by coastal and offshore wetlands [43], eastern lake wetlands, Mon–Xin Lake wetlands, Yunnan–Guizhou Lake wetlands, Qinghai–Tibetan Lake wetlands, northeastern lake wetlands, and marsh wetlands [23], with reference to the results of the existing literature. In the actual calculation of the carbon sink, the same carbon absorption coefficient was selected for 'river wetland' and 'lake wetland', and the same carbon absorption coefficient was selected for 'marsh wetland' and 'artificial wetland'. The same carbon sequestration coefficient was employed for 'swamp wetland' and 'artificial wetland'.

Since different crops, cultivation methods, fertilization conditions, and growth stages of crops can cause variation in the soil respiration of crops, and because farmland systems are subject to strong human influence, the soil respiration of farmland systems in the actual environment may be much larger than the experimentally measured results. The size of soil heterotrophic respiration at limited farmland observation sites in China ranges from

2.48 from t C/hm$^2$·a to 5.93 t C/hm$^2$·a, with an average soil heterotrophic respiration size of 5.43 t C/hm-a$^2$, according to the most recent data of the year [44]. The average heterotrophic respiration of soil in China in 2015 was 3.34 t C/hm$^2$·a [45]. Some scholars listed 14 data from two farmland observation points by consulting the literature and observation station data [44]. In this study, the average value of 14 land heterotrophic respiration data was applied as the soil heterotrophic respiration coefficient of the farmland system employed in the study. This coefficient was confirmed to be 5.26 t C/hm-a$^2$.

In summary, the specific values of the coefficients applied in this study are listed in Table 2:

**Table 2.** Specific values of correlation coefficients for 1 carbon sink calculations.

| Coefficient Name | Specific Values |
| --- | --- |
| Forest | 3.81 t C/hm$^2$·a |
| Garden | 2.38 t C/hm$^2$·a |
| Eastern lakes wetlands | 0.57 t C/hm$^2$·a |
| Yunnan and Guizhou Lakes wetlands | 0.20 t C/hm$^2$·a $^2$ |
| Northeast Lakes Wetlands | 0.04 t C/hm$^2$·a |
| Heterotrophic respiration in agricultural soils | 5.26 t C/hm$^2$·a |
| Grassland | 0.95 t C/hm$^2$·a |
| Coastal and offshore wetlands | 1.73 t C/hm$^2$·a |
| Mengxin Lake wetland | 0.30 t C/hm$^2$·a |
| Qinghai-Tibet Lake wetland | 0.13 t C/hm$^2$·a |
| Swampy wetlands | 0.52 t C/hm$^2$·a |

The crop species selected for the calculation of crop carbon sinks and the correlation coefficients involved in this study were selected as follows [25,33,46] in Table 3:

**Table 3.** Crop carbon sink correlation coefficients.

| Crop | Carbon Absorption Rate | Water Content | Economic Factor |
| --- | --- | --- | --- |
| Rice | 0.4144 | 0.12 | 0.45 |
| Wheat | 0.4835 | 0.12 | 0.4 |
| Corn | 0.4709 | 0.13 | 0.4 |
| Sorghum | 0.45 | 0.13 | 0.35 |
| Grain | 0.45 | 0.12 | 0.4 |
| Potatoes | 0.4226 | 0.7 | 0.7 |
| Beans | 0.45 | 0.13 | 0.34 |
| Cotton | 0.45 | 0.08 | 0.1 |
| Canola | 0.45 | 0.1 | 0.25 |
| Vegetables | 0.45 | 0.9 | 0.6 |
| Peanuts | 0.45 | 0.1 | 0.43 |
| Melon | 0.45 | 0.9 | 0.7 |
| Tobacco | 0.45 | 0.12 | 0.55 |
| Other | 0.45 | 0.12 | 0.4 |

### 3.3. Carbon Sink Projections

NEP is influenced by human activities, natural conditions fluctuate greatly, and there is no certain trend of NEP in China over the years [47]. Some scholars used the IBIS model to predict the total NEP in China from 2015 to 2060 under the conditions of natural variation of the climate [17]. The results show that there was no pattern of variation in NEP values over time. The total NEP in approximately 2020 was not much different from the total NEP in approximately 2060, and the conclusion shows that the average change rate of each type of land area in the study results was less than 1% [17], which provides empirical support for the use of constant NEP coefficients for predicting carbon sinks in this study. Since carbon sinks are influenced by the area of land types, the core of predicting carbon sinks lies in predicting the changes in land use area in China.

Currently, the models employed by scholars in predicting land use change are quantitative models, spatial models, and hybrid models in three major categories. The driving factors involved in each model are mainly human activities and natural impacts. The driving factors may involve only natural influences, such as soil quality, precipitation, and temperature [17], or human activities, such as the GDP growth rate and the population growth rate [48,49]. Some scholars have also considered both human activities and natural conditions to develop a new FLUS model and project the area of each type of land use by 2050 [50]. The predicted results of land use change in this study are derived from a summary of the rate of change of each type of land use area in the relevant literature. The 2030 and 2060 land use areas for each type were then calculated with 2019 national land survey data. Since various empirical studies have shown that land use changes are small, the results obtained by the above approach are still reasonably valid.

*3.4. Data Source*

The energy data and scenario setting indicators employed in the carbon emission measurement and projection were obtained from the China Statistical Yearbook, China Energy Statistics Yearbook, CEADs database, National Bureau of Statistics database, and the National 14th Five-Year Plan Outline. Among them, 2019, 2017, 2012, and annual carbon emission data were derived from the China Carbon Accounting Database.

The carbon neutral years calculated in this study were 2060, 2030, 2019, 2017 and 2012. These years were chosen because 2060 is China's 'carbon neutral' target year and 2028 is the year calculated by the LEAP model as China's 'carbon peak'. The data for 2019 is the most recent land use area data available, while 2017 is the most recent land use area data available in the Statistical Yearbook. As China's land use statistics are updated once every five years, the data for 2012 was chosen. The area of the corresponding land type for 2019 is derived from the 2017 Third National Land Survey, and the data for 2012 was obtained from the China Environmental Statistical Yearbook, China Urban Construction Statistical Yearbook, and China Rural Statistical Yearbook. Since the area of each land type in the China Environmental Statistical Yearbook is calculated based on 'agricultural land', the area of urban greening in each province was determined through the China Urban Construction Statistical Yearbook, and the 'green area' and 'green coverage area' were included. The sum of the 'green area', 'green coverage area', and 'park green area' is considered the garden area to calculate the carbon sink.

Crop production data was obtained from the *China Rural Statistical Yearbook* and the statistical yearbook of each city, in which 'other crops' only contain statistics in the records for some crops, but did not set a specific coefficient of three crops: barley, oats, and buckwheat.

## 4. Results and Discussion

### 4.1. Carbon Emission Results and Projections

In a strong, low-carbon development scenario that is consistent with the national green development strategy, China's future carbon emissions will peak at approximately 10.272 billion tons of $CO_2$ in 2028, which is an increase of 3.75% over 2019, with a compound growth rate of 0.41% of total carbon emissions per year between 2019 and 2028. After peaking in 2028, carbon emissions decline annually to 7227 million tons in 2060, which is a decrease of 29.77% from 2028. The compound rate of decline in total carbon emissions is approximately 1.1% per year between 2028 and 2060, which is approximately 2.7 times the growth rate of carbon emissions between 2019 and 2028 in Figure 2.

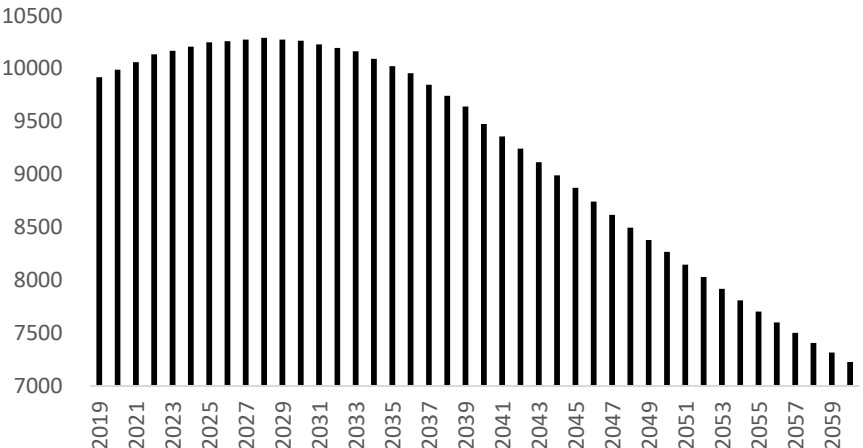

**Figure 2.** 2019–2060 LEAP Forecast China Carbon Emissions (Million tons $CO_2$).

The projections of carbon emissions in this study are generally consistent with those of other authors. The academic opinion is that with a GDP growth rate of 5.0% and a non-fossil energy share of 23.0% and above, China's carbon emissions will peak between 2025 and 2029, with a peak value of 11.2 billion tons on average and a distribution range of 10.5–11.9 billion tons [51]. When the share of non-fossil energy is set at 26.5% and the GDP growth rate is 5%, carbon peaks in 2026 and reaches a peak of 10.5 billion tons [51], which is almost consistent with this study.

### 4.2. Carbon Sink Results and Projections

The 2060 and 2028 land use areas of each type refer to the existing literature on predicting future land use changes in China and summarize the research findings of relevant scholars, as well as China's actual development policies, such as the intensive use of regional land resources. In this study, we summarized the existing literature on predicting future land use changes in China and the actual development policies in China, such as the intensive use of regional land resources and the linkage between urban construction land and rural construction land [51]. This study holds that under the strategic scenario of China's adherence to low-carbon sustainable development, forestland, grassland, garden land, cultivated land, and wetland will increase by 1.38%, 0.93%, 0.31%, 1.09%, and 1.04%, respectively, in 2060 compared with 2019 [17,48–50].

The land use area for each year is shown in Table 4.

**Table 4.** Land use by year (million square hectares).

|  | WOODLAND | GRASSLAND | GARDEN | ARABLE LAND | WETLANDS |
| --- | --- | --- | --- | --- | --- |
| 2060 | 28,806.10 | 26,699.02 | 2023.41 | 12,924.92 | 5460.85 [1] |
| 2028 | 28,510.97 | 26,514.51 | 2018.72 | 12,820.87 | 5460.67 [1] |
| 2019 | 28,412.59 | 26,453.01 | 2017.16 | 12,786.19 | 5404.64 [1] |
| 2017 | 25,280.19 | 21,932.03 | 2117.44 | 13,488.12 | 5360.26 [1] |
| 2012 | 25,339.7 | 21,956.5 | 2016.7 | 13,515.8 | 3848.55 [1] |

[1] Wetlands is the sum of wetland and water area in the "three national land surveys".

We calculated the land use area for future years by combining the projections of other scholars on future land use area, which methodologically ensures that the calculation results are comparable with similar studies. As the annual changes in land use are relatively insignificant, the differences between various studies in predicting future values are minimal, and the direct use of other scholars' research results does not affect the conclusions of this study. It is impossible to accurately predict the area of various wetlands in the predicted land use in 2028 and 2060, and there is a lack of prediction data of the output of various types of crops in the calculation of the carbon sink in 2028 and 2060. Therefore,

the NEP coefficient used by wetlands is the average NEP coefficient of wetlands in 2019 of 0.59 t C/hm$^2$, and the NEP coefficient used by cultivated land is the evaluation NEP coefficient of cultivated land in 2019 of 0.60 t C/hm$^2$.

In terms of carbon sinks generated by various types of land, forestland is the highest, with an average annual carbon sequestration of 1.039 billion tons C, and the carbon sinks of forestland increase year by year. In addition to forestland, grassland is the second largest land use type of carbon sink. The total amount of carbon sinks is approximately 235 million tons C per year, which is on the rise. The total amount of grassland carbon sinks will reach 254 million tons of carbon dioxide in 2060. The total annual carbon sinks of gardens, wetlands, and cultivated land are 45 million tons C, 32 million tons C, and 83 million tons C, respectively.

It is worth mentioning that the total carbon sinks of cultivated land decreased gradually from 2012 to 2019. Although the growth of crops in the farmland ecosystem will cause large NPP, the NEP value of the whole farmland ecosystem is small due to the large heterotrophic respiration of the land. From 2012 to 2019, although the crop yield gradually increased, thus increasing the total NPP of crops, the cultivated land area also increased, and the increase in heterotrophic respiration caused by the increase in cultivated land area was greater than the added value of NPP. Thus, the carbon sink of the farmland system decreased from 2012 to 2019. China is a large agricultural country with a large crop yield and wide farmland area. We should focus on the use of scientific farming methods and environmentally friendly feed to enhance the carbon sink capacity of the farmland system. In this study, the same NEP coefficient as that in 2019 was applied in the prediction of cultivated land carbon sequestration in 2028 and 2060. However, if agricultural technology advances, the reduction in land heterotrophic respiration and the increase in crop yield can be simultaneously realized. If the land heterotrophic respiration coefficient in 2060 becomes 80% of that in 2019 and the crop yield per unit area increases by 20% compared with that in 2019, the crop carbon sequestration in 2060 can reach 300 million tons C, which is approximately four times the current level and even higher than that of grassland in 2060 in Table 5.

**Table 5.** Carbon sinks by land use type (million t C).

|      | WOODLAND   | GRASSLAND | GARDEN  | WETLANDS | ARABLE LAND |
|------|------------|-----------|---------|----------|-------------|
| 2060 | 109,751.24 | 25,364.07 | 4815.72 | 3154.48  | 7813.51     |
| 2030 | 108,626.78 | 25,188.78 | 4804.55 | 3166.25  | 7750.61     |
| 2019 | 108,251.97 | 25,130.36 | 4800.84 | 3170.17  | 7729.65     |
| 2017 | 96,317.52  | 20,835.43 | 3382.98 | 3574.28  | 9037.46     |
| 2012 | 96,544.26  | 20,858.68 | 4799.76 | 2800.72  | 9373.10     |

The contribution of all kinds of land use to carbon sinks is also basically stable. Forestland contributed 72.73% of the total carbon sequestration in 2060. The grassland carbon sink accounts for 16.34% of the total carbon sink on average, and the proportion of the grassland carbon sink also increases steadily year by year. From 2012 to 2060, the proportion of the grassland carbon sink in the total carbon sink increased from 15.52% to 16.81%. On average, the total carbon sinks of forestland and grassland accounted for 88.77% of the total carbon sinks, which can be determined to make the greatest contribution to ecosystem carbon sinks. On average, garden carbon sinks account for approximately 3.15% of the total ecosystem carbon sinks, wetland carbon sinks account for approximately 2.22% of the total ecosystem carbon sinks, and cultivated land carbon sinks account for approximately 5.86% of the total ecosystem carbon sinks in Table 6.

**Table 6.** Proportion of carbon sinks for various land use types.

|  | WOODLAND | GRASSLAND | GARDEN | WETLANDS | ARABLE LAND |
|---|---|---|---|---|---|
| 2060 | 72.71% | 16.81% | 3.19% | 2.10% | 5.18% |
| 2030 | 71.99% | 16.69% | 3.18% | 2.10% | 5.14% |
| 2019 | 71.74% | 16.65% | 3.18% | 2.10% | 5.12% |
| 2017 | 63.83% | 13.81% | 2.24% | 2.37% | 5.99% |
| 2012 | 63.98% | 13.82% | 3.18% | 1.86% | 6.21% |

Since the land use area data in 2060 and 2028 are derived from the forecast, the relative growth rate of the carbon sink is highly correlated with the predicted value of the land use area in 2060 compared with that in 2019. The data of 2012, 2017, and 2019 were derived from historical real data. An analysis of the data growth rate in these three years revealed that in the short term, the changes in various types of land use carbon sinks fluctuate violently and inconsistently due to the changes in various types of land use areas in the short term. In the long term, however, the areas of various types of land show a stable and slight growth trend, which explains why the total amount of carbon sequestration has maintained stable growth in the long run in Table 7.

**Table 7.** Carbon sink growth rate of various land use types (compared with the previous representative year).

|  | WOODLAND | GRASSLAND | GARDEN | WETLANDS | ARABLE LAND |
|---|---|---|---|---|---|
| 2060 | 1.04% | 0.7% | 0.23% | −0.37% | 0.81% |
| 2030 | 0.35% | 0.23% | 0.08% | −0.12% | 0.27% |
| 2019 | 12.40% | 20.61% | 41.91% | −11.30% | −14.47% |
| 2017 | −0.23% | −0.11% | −29.52% | 27.62% | −3.58% |
| 2012 | 1.04% | 0.7% | 0.23% | −0.37% | 0.81% |

In fact, there are many factors that affect the carbon sink capacity of an ecosystem, such as climate change and plant type, but it is difficult for us to precisely predict the changes in the distribution area of various plants over time. As previously mentioned, there is no uniform methodology for carbon sink calculations in academia, fluctuations in NEP coefficients will be large, and it is not operationally feasible to accurately estimate specific values for specific future years. Although there are shortcomings in our use of such a method to calculate carbon sinks, it is consistent with mainstream thinking in academia in terms of methodology, data sources, and calculation theory, and does so in a way that makes the results of the study comparable.

*4.3. Carbon Gap*

Through the calculation and prediction of carbon emissions and carbon sinks, China's carbon accounts in 2012, 2017, 2019, 2030 and 2060 can be obtained; they are presented as follows in Table 8 and Figure 3:

**Table 8.** Carbon accounts for 4 years (million tons of $CO_2$).

|  | TOTAL CARBON SINK | TOTAL CARBON EMISSIONS | NET EMISSIONS | ANNUAL VARIATION | CARBON SINKS/ CARBON EMISSIONS |
|---|---|---|---|---|---|
| 2060 | 5534.75 | 7226.70 | 1691.95 | −3.21% | 76.59% |
| 2030 | 5484.37 | 10,262.10 | 4777.73 | 1.28% | 53.44% |
| 2019 | 5466.37 | 9753.59 | 4287.22 | −1.13% | 56.04% |
| 2017 | 4711.15 | 9097.27 | 4386.12 | 0.41% | 51.79% |
| 2012 | 4927.14 | 9224.31 | 4297.17 | - | 53.41% |

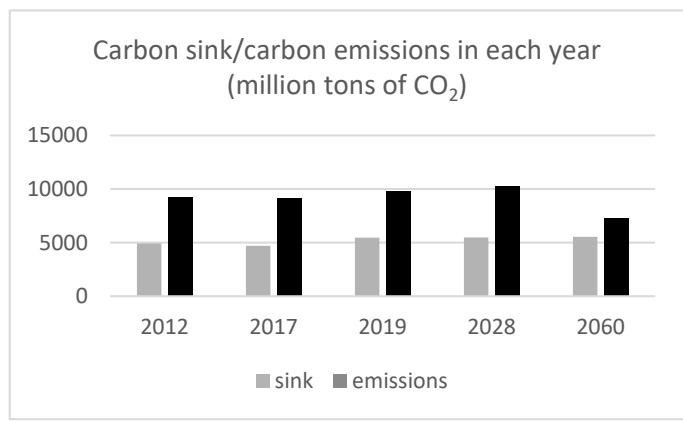

**Figure 3.** Carbon sink/carbon emissions in each year (million tons of $CO_2$).

Presently, China's carbon sink/carbon emissions are approximately 55%. In the past few years, China's net emissions have shown a stable growth trend. The net emissions have increased at a compound growth rate of 1.28% since 2019, reaching a peak in 2028, and reaching a maximum deficit of approximately 4.8 billion tons of carbon dioxide in the carbon account.

After the carbon peak, with a decline in carbon emissions, net emissions also decreased at an average annual rate of 3.21%, which is much higher than the average annual rate of 1.1% of carbon emissions. Although the total amount of carbon sink has not increased at a high rate, with the simultaneous force at both ends of the "sink and source", after the carbon peak, the rate of net carbon emission reduction is much higher than that of the carbon emission reduction. However, there was still a net emission of 1.692 billion tons of $CO_2$ in 2060, and the carbon sink/carbon emission was 76.59%, indicating that the carbon sink of the ecosystem alone cannot completely neutralize the carbon dioxide generated by China's energy consumption. By relying solely on the ecosystem carbon sink, there were still approximately 1.7 billion tons of net carbon dioxide emissions in 2060, accounting for 23.41% of the total carbon emissions in the same period.

We have not taken into account changes in carbon sink capacity when calculating carbon sinks, and there is a good chance that if more scientific planting methods were used to increase carbon sinks in each ecosystem, we would be closer to the goal of 'carbon neutrality' by 2060. Unfortunately, we cannot analyze this scenario due to a lack of data to support it.

## 5. Conclusions

According to the LEAP model, China's carbon emissions will increase at an annual rate of 0.41% to reach a peak in 2028, and the peak value will increase by 3.75% compared with that in 2019. The annual carbon emissions will then decrease at an annual rate of 1.1% to approximately 7.2 billion tons of carbon dioxide in 2060. The carbon emissions will decrease by 30% compared with that in 2030 and by 27% compared with that in 2019.

In terms of the long-term trend, the land use area of various types in China has increased at a small rate, and the stable increase in the area has ensured a stable increase in the total carbon sink of China's ecosystem. In 2060, the total carbon sink reached 5.5 billion tons of carbon dioxide, accounting for 76.59% of the carbon emissions in the same period.

Forestland and grassland are the main contributors to the ecosystem carbon sink; their total carbon sink accounts for approximately 88.77% of the total ecosystem carbon sink. Over time, forestland carbon sinks and grassland carbon sinks account for an increasing proportion of the whole ecosystem carbon sink. By increasing the urban greening area, we can not only improve people's environment, but also increase the carbon sink in the ecosystem. In addition to forestland and grassland, cultivated land is the third largest contributor to the ecosystem carbon sink, mainly due to the large area of cultivated land

in China. The farmland system is greatly affected by human beings and different crops, farming methods, and fertilizer types, which will affect NEP. Therefore, the NEP coefficient of the farmland ecosystem is relatively unstable. Advanced farming methods may produce an immense carbon sink in the farmland system, but may also transform it from a carbon sink into a carbon source. Therefore, China must pay attention to the carbon sink generated by the farmland ecosystem in the future to ensure that the farmland ecosystem scientifically and reasonably becomes a powerful carbon sink contribution source.

Despite the significant decline in carbon emissions after our carbon peak, there will still be net emissions of about 1.7 billion tons of $CO_2$ in 2060, based on current development patterns. Only through the ecosystem does the carbon sink not completely neutralize the carbon emissions generated by energy consumption. Since forestland contributes the largest carbon sink, focusing on increasing the area of forestry plantations and increasing the proportion of trees in parks can both increase the total carbon sink. Since arable land is prone to interconversion between carbon source—carbon sink, blind expansion of arable land area may make the farmland ecosystem a huge carbon source. Therefore, the development of agriculture requires the improvement of major agricultural technologies, reasonable fertilizer use, and scientific plant cultivation to ensure that the carbon sink produced by crops is greater than the heterotrophic respiration of farmland soils, so that farmland ecosystems can play the role of a carbon sink. China has a long way to go for carbon neutrality by increasing the use of new energy while protecting the ecological environment. By completing the transformation of low-carbon development of various industries through new technologies, China's strategic goal of carbon neutrality can be expected to be achieved in 2060.

**Author Contributions:** Conceptualization, N.W. and T.S.; Data curation, Y.Z., E.W. and S.D.; Formal analysis, Y.Z. and X.Z.; Funding acquisition, N.W. and T.S.; Methodology, Y.Z.; Project administration, T.S. All authors have read and agreed to the published version of the manuscript.

**Funding:** This research was funded by two projects. One was funded by the National Social Science Fund of China grant number [19BJL046]. The other was funded by Chinese Research Academy of Environmental Sciences grant number [OITC-G190270565]. And The APC was funded by Chinese Research Academy of Environmental Sciences grant number [OITC-G190270565].

**Institutional Review Board Statement:** Not applicable.

**Informed Consent Statement:** Not applicable.

**Data Availability Statement:** The data comes from: https://data.cnki.net.

**Conflicts of Interest:** The authors declare no conflict of interest.

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
