# Peer review of "Accounting for China’s Net Carbon Emissions and Research on the Realization Path of Carbon Neutralization Based on Ecosystem Carbon Sinks"

_sustainability, doi:10.3390/su142214750_

Round 1

Reviewer 1 Report

In this paper, the authors predicted the carbon emissions and carbon sinks for seeking the potential path to achieving carbon neutrality. The study finds that the carbon emissions will peak in 2028 with the amount of 10.27 billion tons. The amount of carbon sinks remained stable, which is not sufficient for carbon neutrality. The authors find several results which could possibly contribute to the existing literature. The research topic has marginal potentials for practical implications, especially for the current green development strategies in China. However, I believe the paper could improve if the authors take the following issues into consideration during the revision process.

(1) the introduction part should be further improved. the motivation of this paper is not clearly stated in this part. It is suggested to reorganize the structure of this part and describe the objective of this study. Moreover, the author should also highlight its novelties in the introduction part. What is the contribution of this study? The method and the data are widely used by other scholars. What is the difference between this study and others?

(2) The literature review should be enhanced. Compared with 2.1 and 2.2, the literature on carbon sink calculation is too short.

(3) I suggest the authors to provide more justification for the selection of key variables in Table 1. It is also necessary to explain why you choose LEAP model as well as the carbon sink measurement model.

(4) As the author mentioned, the five representative years of 2012, 2017, 2019, 2030, and 2060 are selected to predict the ecosystem carbon sinks. It is better to explain why you choose the five years.

(5) The policy implications are common sense. It is not clearly linked to what you find in the above analysis. It is better to propose the suggestion accordingly with the empirical findings.

(6) The language needs to be further polished.

Author Response

Thank you very much for your criticism and suggestions. I have revised the introduction, literature review and policy recommendations according to your suggestions, and explained the reasons for selecting the five representative years.

LEAP model is a commonly used software to simulate energy consumption under different scenarios. The key variables in the model are derived from relevant literature and model simulation. Since the future values of key variables cannot be accurately determined, they can only be assumed according to the results of relevant literature. The assumptions in this paper are basically consistent with those of other scholars.

Please refer to the attachment for the revised version. Think you again.

Reviewer 2 Report

本文主要研究了中国净碳排放量的核算以及基于生态系统碳汇的碳中和实现路径。通过构建LEAP-China模型,本文预测了2019年至2060年的碳排放量,发现中国的碳排放量将以每年0.41%的速度增长,并在2028年达到峰值。峰值将比2019年高出3.75%。从那时起,它每年萎缩1.1%。在相关文献回顾的基础上,作者预测了2012年、2017年、2019年、2030年和2060年的生态系统碳汇,计算了中国这五个关键年的净碳排放量,定量分析了中国实现碳中和的路径。作者的结果表明,土地利用产生的生态系统碳汇更加稳定。2060年,中国碳汇总量达到55亿吨,占同期碳排放量的76.59%。路径分析表明,林地和草地是生态系统碳汇的主要贡献者,其总碳汇占生态系统总碳汇的88.77%。

本研究结果为我国碳中和提供了碳峰预测,在总结大量相关研究的基础上,从碳汇的角度分析了我国碳中和的实现路径。笔者认为,本文丰富了我国对碳中和碳汇的研究,但研究深度有待进一步提高。对于本文,我有以下问题和建议:

(1)在碳排放预测部分,能否与实际碳排放数据与其他学者的碳峰预测进行比较,验证本文作者的碳峰预测结果?

(2)碳汇预测结果能否可视化,以增强文章的阅读能力?

Translation:

This paper focuses on the accounting of net carbon emissions in China and the pathway to carbon neutrality based on ecosystem carbon sinks. By constructing the LEAP-China model, this paper predicts carbon emissions from 2019 to 2060 and finds that China's carbon emissions will grow at an annual rate of 0.41% and peak in 2028. The peak will be 3.75% higher than in 2019. From then on, it shrinks by 1.1% per year. Based on a review of relevant literature, the authors project ecosystem carbon sinks in 2012, 2017, 2019, 2030, and 2060, calculate net carbon emissions in China for these five key years, and quantitatively analyze China's pathway to carbon neutrality. The authors' results show that ecosystem carbon sinks from land use are more stable. in 2060, China's total carbon sink reaches 5.5 billion tons, accounting for 76.59% of carbon emissions in the same period. The pathway analysis shows that woodlands and grasslands are the main contributors to ecosystem carbon sinks, and their total carbon sinks account for 88.77% of the total ecosystem carbon sinks.

The results of this study provide carbon peak projections for carbon neutrality in China, and analyze the pathways to achieve carbon neutrality in China from the perspective of carbon sinks on the basis of summarizing a large number of relevant studies. I believe that this paper enriches the research on carbon neutral carbon sinks in China, but the depth of the research needs to be further improved. For this paper, I have the following questions and suggestions.

(1) In the part of carbon emission prediction, can the actual carbon emission data be compared with the carbon peak prediction of other scholars to verify the carbon peak prediction results of the authors of this paper?

(2) Can the carbon sink prediction results be visualized to enhance the readability of the article?

Author Response

评审专家您好!非常感激您的建议,我已经按照您的建议进行了修改。请详见修改后的文件。

Translation:

Dear reviewer:

Thank you very much for your suggestions, I have made changes according to your suggestions. Please see the revised manuscript for details.

Reviewer 3 Report

The theme is topical, but several concerns must be solved before re-evaluating the manuscript and possibly considering it for publication. The paper is overall well written and well presented. Please answer point to point on these recommendations.

The introduction provides a brief background and sets the scene for what the reader might expect from this study. However, the authors must clearly formulate the study's aims in the Introduction section.

The way in which the paper is written indicates that the authors had a clear indication of the direction of the research from the outset. An element of bias can be felt throughout. Kindly articulate the research gap and clearly outline the contribution made through this study to academia. The contribution to the literature needs to be articulated in a better way.

The reference list must be formatted according to the requirements of the Sustainability journal.

Author Response

Thank you very much for your criticism and suggestions. I have revised the introduction and references according to your suggestions.

The revised version has been uploaded as an attachment. Please check.

Round 2

Reviewer 1 Report

After the careful revision, the revised manuscript is improved. The authors addressed my comments well. 

However, 

(1) it still have some minor typos.e.g., Line 77, it has a space missing. Similar mistakes can also be fond in Line 89. it is better to check the writing style carefully.

(2) Moreover, it is better to explain the abbreviations when they appear first time, e.g., CCUS, BECCS, etc.

(3) the language needs to be better polished. the revised sentences in red are not native. e.g., line 137-138. line 142-145.

Author Response

Thanks again for your comments, I have rechecked the manuscript and made changes based on your comments